# Inter-annual sea ice thickness variability in the Bay of Bothnia

Iina Ronkainen[1], Jonni Lehtiranta[1], Mikko Lensu[1], Eero Rinne[1], Jari Haapala[1], Christian Haas[2]

[1]Finnish Meteorological Institute, Helsinki, 00560, Finland
[2]Alfred Wegener Institute, Bremerhaven, 27570, Germany

*Correspondence to*: Iina Ronkainen (iina.ronkainen@fmi.fi)

**Abstract.** While variations of Baltic Sea ice extent and thickness have been extensively studied, there is little information about drift ice thickness, distribution and its variability. In our study, we quantify the inter-annual variability of sea ice thickness in the Bay of Bothnia during the years 2003-2016. We use various different data sets: official ice charts, drilling data from the regular monitoring stations in the coastal fast ice zone and from helicopter- and ship-borne electromagnetic soundings.
We analyze the different data sets and compare them to each other to characterize the inter-annual variability, to discuss the ratio of level and deformed ice, and to derive ice thickness distributions in the drift ice zone. In the fast ice zone the average ice thickness is 0.58 +/- 0.13 m. Deformed ice increases the variability of ice conditions in the drift ice zone where the average ice thickness is 0.92 +/- 0.33 m. On the average, fraction of deformed ice from total volume is 50 to 70%. In heavily ridged ice regions near the coast, mean ice thickness is approximately half a meter thicker than in pure thermodynamically grown fast
ice. Drift ice exhibits larger inter-annual variability than fast ice.

## 1 Introduction

The Baltic Sea belongs to the seasonal sea ice zone, extending from 54° N to 66° N, and has a total area of  422 000 km$^2$. According to the classification of the Finnish Ice Service, the Baltic ice season is mild when ice extent is below 115 000 km$^2$, severe when the extent is above 230 000 km$^2$ and extremely severe when it is above 345 000 km$^2$. The last occurrence of
almost complete ice cover was in 1987 (96%, Vihma and Haapala, 2009). Other major seas that are seasonally ice covered include the Sea of Okhotsk and Bohai Sea in Asia and Hudson Bay and Gulf of St. Lawrence in North America.

Bay of Bothnia is the northernmost basin of the Baltic Sea north of the sound of Quark at 63.5° N. It is a semi-enclosed basin with a length of about 300 km, width of 100-150 km and an area of approximately 36 000 km$^2$. The average depth is 41 m and
maximum depth 146 m (Leppäranta and Myrberg, 2009). During the last 100 years the Bay of Bothnia has not frozen completely over only during extremely mild winters 2014/2015 for sure and most probably also in 1929/1930 (Uotila et al., 2015). The width of the fast ice varies with the sheltering of the archipelago and in more open areas its boundary follows roughly the 10 m depth contour. In the drift ice zone, the flux of ice trough the 25 km wide passage of Quark is small and has only a minor effect on the ice mass balance of the basin which is mostly determined by the thermodynamic and dynamic

processes within the basin itself, i.e. according to our understanding the exchange of ice with the Gulf of Bothnia in the south is negligible.

In the fast ice zone, ice grows thermodynamically because the ice is attached to the coast and does not move. Along the coastline of Bay of Bothnia there is a large area where the shallowness of the sea allows fast ice to form every winter. Depending on the ice thickness, the boundary of the fast ice zone is where the depth is 5-15 m. On the contrary, in the drift ice zone, ice floes move along the currents and winds and pile up to ridges. Ridges can grow up to several meters. When ice is piling up in one region, open water forms in another, and favorable conditions for formation of new ice turn out. Thus, the growth of ice in the drift ice zone consist of thermodynamic and dynamic processes (Leppäranta and Myrberg, 2009).

Forcings in the thermodynamic and dynamic processes are different. Sensible and long wave radiation fluxes and snow accumulation are the main factors to drive thermodynamic growth. The thicker the ice grows the more it insulates itself and grows slower. Snow layer on the ice acts also as insulation. On the other hand, if there is lots of snow on the ice, sea water floods on the ice and can form snow-ice on top of the ice cover. Dynamic processes are more complicated than thermodynamic ones. Ice motion is driven by winds and currents and can form rafted, ridged or brash ice (Leppäranta and Myrberg, 2009).

The fast ice zone of the Bay of Bothnia consists of level ice types which in more sheltered areas grow purely thermodynamically. In the beginning of ice season, there is fast ice in more sheltered areas. During the winter the fast ice zone expands to shallow drift ice areas, where depth is under 10 m. As the fast ice boundary advances offshore during the ice season the ice types in more exposed areas may drift and experience deformations. This initial thickness variation is smoothed by thermodynamic growth after the ice stabilizes into fast ice. The drift ice zone is a mixture of level and deformed ice types of different thicknesses, and typically has a high ridge density. In heavily ridged areas the largest ridges may reach 20 m below the surface (Palosuo, 1975). The variability of thermal and dynamic forcing leads to large ice thickness variations both in space and time. In addition there are large coastal gradients generated by coastal process cycles like lead opening, refreezing and deformation (c.f. Pärn and Haapala, 2011, Oikkonen et al, 2016).

Sea ice conditions are typically characterized by ice extent, ice thickness and ice concentration. Baltic ice charts, providing the ice covered area, have been drawn and the phases of the ice season have been followed systematically since 1915. In the Bay of Bothnia there is a more than 100 years long record of drill-hole measurements and of the dates of freeze up and melt from fixed sites. These have been restricted to the fast ice zone. The thickness of level ice types within the drift ice zone is measured or estimated by icebreakers and reported in the ice charts. These values usually seek to characterize the regional level ice thickness for navigation purposes and have limited accuracy. The thickness of deformed ice has been studied in individual campaigns: from thickness profiles determined by air- and ship-borne electromagnetic (EM) sounding since 2003 and indirectly from surface profiles since 1988. These data sets do not cover all years, and the individual campaigns do not cover

all parts of the Bay of Bothnia. Also, most of the campaigns have been conducted during the midwinter period from February to March, when the ice may still grow and deform further.

Previous studies of the long term thickness statistics in the Bay of Bothnia have mostly concentrated on annual maximum level ice thickness in the fast ice zone, especially at Kemi which is close to northernmost end of the basin. Leppäranta and Seinä (1985) determined that there was a statistically significant increasing trend in the maximum annual ice thickness at Kemi from 1912 to 1984. An increasing trend at Kemi was also recorded in later research from 1912 to 1996 (Jevrejeva et al., 2004) and from 1912 to 2011 (Haapala et al., 2015). The mean maximum annual ice thickness at Kemi from 1912 to 2011 is 73 cm (Haapala et al., 2015).

In the Bay of Bothnia or even in the Baltic Sea, the inter-annual changes of drift ice thickness have not been studied before. However, many similar studies have been carried out in the Arctic (Meier et al., 2014) and Antarctic. Airborne EM measurements have been used to determine reduction in ice thickness near the North Pole 1991-2007 (Haas et al., 2008), to describe the ice thickness in the Northwest Passage (Haas and Howell, 2015), to estimate ice thickness in a pan-Arctic survey (Haas et al., 2010) and to study ice thickness in the Barents Sea (King et al., 2017). Air and ship-borne EM measurements have also been carried out in the Antarctic (Haas, 1998; Weissling et al., 2011; Sugimoto et al., 2016). In seasonal seas, ice thickness has been studied for example by upward-looking sonar in the Bohai Sea (Su et al., 2013).

The inter-annual variability of ice extent generally follows the variations in large-scale atmospheric forcing. In the Baltic Sea the North Atlantic Oscillation (NAO) index is a good proxy for this. Strong westerlies blow over the North Atlantic during positive phases of the NAO index and conditions over the Baltic Sea are mild and moist. During negative NAO phases westerlies weaken or are blocked totally and then winters are more severe. Vihma and Haapala (2009) found that during the winters with strongly positive NAO index the average of maximum annual ice extent in the Baltic Sea was 121 000 km$^2$, whereas during the winters with strongly negative NAO index it was 259 000 km$^2$. However, for example in 1985-1986 the NAO index was negative only in February, while the seasonal NAO index was positive, and the ice extent was very large in that winter. The influence of NAO to ice thickness has not been studied in the northern Baltic Sea. Specifically, it is not known how the different wind and temperature statistics, characteristic to positive and negative NAO, combine to generate a specific ratio of level and deformed ice types in the ice volume budget.

It is important to investigate sea-ice changes in the seasonal sea ice zone for two reasons. First, for hundreds of years, there has been interest in sea-ice observations to help navigation. Second, interest in climate change is increasing, and sea-ice changes strongly reflect the changing climate.

The average ice season in the Bay of Bothnia lasts from November to May (Haapala et al., 2015). Thus, sea ice is an essential factor in the area. Every winter, both Finnish and Swedish ice breakers assist ships on route to harbours in Bay of Bothnia. Winter navigation has continuously increased and more information on the ice conditions is required. Especially ridged areas can cause major problems to ships. Milder winters are not necessarily easier for navigation. The ice thickness distribution and the amount of ridges affect navigation more.

In this study we are using all available measurements from various helicopter and shipborne EM field campaigns to determine the variability of sea ice thickness in the Bay of Bothnia in more detail. The aim of our study is to combine and compare these data with existing in situ drilling data from the fast ice zone and ice chart thickness information. Despite the fact that the EM measurement haven't conducted along same paths or areas in every years, we presume that these unique data set is sufficiently extensive to reveal main features of regional and inter-annual variability of ice thickness. The data has been used in developing satellite sea ice retrieval methods (Gegiuc et al, 2018) and validation of numerical models (Löptien et al, 2013; Pemberton et al, 2017), but only geophysical analysis of the data has been conducted only by Haas (2004a) who concluded that level ice thickness was close to the values reported in ice charts but the actual mean ice thickness is much larger due to ridged ice. In this paper, we estimate, for the first time, the variability of ice thickness in the drift ice region and generate ice thickness distributions for individual years as well as discuss its inter-annual variability. We also provide an estimate of the climatologically averaged thickness distribution for the Bay of Bothnia.

## 2 Data and methods

Ice thickness can be measured by several methods: visual observations from ships, drilling, electromagnetic induction (EM) sounding from sledges, ships, or aircraft, upward-looking sonar from moorings or submarines and satellites altimetry (Eicken et al., 2009; Haas, 2017). We use several different data sets to get the best possible overview of the variability of sea ice thickness. Ice charts divide the ice cover in regions representing ice fields with different characteristics and age, and the thickness values seek to represent the typical thickness of level ice in each region. Drilling data at fixed observation stations are accurate and long term but have been made only in the fast ice zone. The EM data resolve the detailed ice thickness distributions in the drift ice zone, but study regions and times have so far been limited to few campaigns during the maximum ice extent.

To define the sea area, we used the definition of the southern boundary of the Bay of Bothnia (Leppäranta and Myrberg, 2009): Iskmo-Raippaluoto-Björkö-Lappören-Valassaaret-Hadding Peninsula (see Fig. 1). The area of the Bay of Bothnia is 36 260 km$^2$. We mark the winters according to the year in the spring. For example 2003 means the winter 2002-2003.

## 2.1 Ice charts

During ice winters, ice charts are published daily by the ice service of the Finnish Meteorological Institute. At the very beginning and end of the season, frequency of the charts is usually reduced to bi-weekly maps. Apart from graphic charts gridded versions with different resolutions are prepared for various purposes. These include grids for concentration, average thickness, maximum thickness, minimum thickness, deformation numeral (degree of ice ridging), and sea surface temperature. The deformation numerals from 0 to 5 denote level ice, rafted ice, slightly ridged ice, ridged ice, heavily ridged ice, and brash barrier. The graphic charts, on the other hand, represent these types with qualitative symbols only.

The main information sources for the charts are satellite images for ice existence and concentration and in-situ measurements and visual observations for ice thickness. The thickness values are based on observations made by the crew of icebreakers and, for nearshore fast ice, at fixed stations. The station values are from drilling while the icebreaker values are estimates from tilting floes during transit in ice and occasional drillings. The thickness values refer to ice types with a flat surface that can be level ice or rafted ice and seek to characterize the regional conditions. The deformation numerals are based mostly on visual icebreaker observations and seek to be a regionally representative description of the conditions from the point of view of navigational difficulty. No rules for estimating total thickness from ice chart thicknesses and the numerals have been established yet although a clear correlation exists (Gegiuc et al., 2018).

We calculated the average annual statistics for the Bay of Bothnia using gridded ice charts for the 14 season period 2003-2016. The grid resolution is 1/60 of a degree in NS direction and 2/60 degree in EW direction. True grid cell areas were used in calculations and the effect of land point occurrence in the cells was estimated. The common period was chosen to be from 1st October to 31st May. For ice-free days grids with zero values for ice parameters were used. During the very early phases of ice season the charts are bi-weekly and the latest ice chart was assumed to stay valid during the intermediate days. The same was assumed for the few gaps, usually one day in duration, during the main ice season. Due to that this is mainly done for the very early phases of ice season, when there is not much ice yet, this barely affects our results.

## 2.2 EM data

One of the cost efficient ways to measure sea ice thickness over a large area is electromagnetic induction (EM) sounding. EM sounding allows sea ice thickness measurements from a moving platform. Here we use data from a helicopter-towed EM-bird (HEM, Haas et al., 2009) and ship-borne Geonics EM-31. EM instruments induce eddies of alternating electric current in conductive layers in the underground and measure the amplitude and phase of the resulting EM fields. From these, the distance to the different layers can be accurately determined (e.g. Haas et al., 1997). With sea ice thickness measurements, the seawater is assumed to be a half-space of constant, known electrical conductivity underneath the resistive sea ice, and the distance to the ice-water interface can be retrieved with a single measurement frequency. EM measurements in the brackish water of the

Baltic Sea depend on in situ measurements of local seawater conductivity. When the EM instrument is suspended from a moving vehicle, the distance to sea ice is not fixed and needs to be measured separately by a laser distance sensor and subtracted from the EM-retrieved distance to seawater to get ice thickness.

A single-frequency EM device cannot differentiate between snow and ice because they are both highly resistive. Therefore snow thickness is always included in the measured total ice thickness. Similarly, conductive layers like flooded snow on ice or porous ridge keels cannot be distinguished but will lead to underestimates of total ice thickness.

The EM thickness measurements are weighted averages over the instrument footprint. The footprint size depends on the
distance between the instrument and the ice-water interface. In addition, EM measurements are affected by smoothing over the footprint, i.e. the area from which an EM device receives most of the return signal. It is roughly five meters for the ship-borne EM-31 and between 3 to 4 times the flying altitude for HEM bird. Typical altitude for the HEM bird is about 15 m, resulting into a footprint size in the order of 50 m. Thickness variations on scales smaller than the footprint are smoothed and therefore ridge keel depths are underestimated or closely arranged keels may join into one signature. This affects especially
the tail part of the thickness histogram. Additionally the instrument sensitivity decreases with increasing distance to sea water, and in general thicknesses greater than 4 meters are seen as unreliable. Usually the EM thicknesses are accepted at their face value and interpreted as average thickness over footprint.

Results of ice thickness surveys are commonly displayed as thickness distributions (histograms). Caution is due when
interpreting and comparing thickness histograms and their mean values. First, thickness refers to the distance between the ice surface and approximate ice-water interface. Considerations involving ice mass balance, especially the transition of level ice types to ridged ice, must take into account the relative void content (porosity) which can be 20-40% for the unconsolidated part of ridge keels (Leppäranta, 2005). We estimate that we underestimate ridge keel volume by 4 % due to this effect, assuming an exponential distribution model for keel depths.

To present the distribution of ice thickness we computed normalized frequency histograms with a bin width of 0.1 m from EM data. In some years the spatial density of EM surveys is much higher in certain areas than elsewhere, especially close to the land base of EM flights. To avoid bias from the dominance of such areas we first calculated histograms in 1 NM grid for each grid cell and then averaged all histograms from the grid cells to one histogram. In addition to histograms pertaining to each
campaign, a histogram for all five years of helicopter EM data was constructed by averaging the annual histograms.

### 2.2.1 Helicopter EM data

HEM surveys were carried out in the Bay of Bothnia in years 2003, 2004, 2005, 2007 and 2011 (Table 1). The flights were mostly made during a short time period from the end of February to the first half of March, i.e. potentially before the ice

reached its maximum annual thickness. The measurements are from different routes in different years. Thus, comparison of the measurements in different years is not unproblematic.

Haas (2006) has shown that ice thicknesses are overestimated in brackish water shallower than 15 to 20 m with the signal frequency of 4 kHz used here. This is due to induction in the seafloor with its lower conductivity compared to seawater. To prevent this error, we masked the data when water depth was less than 15 m.

The first HEM measurements in the Baltic Sea were conducted within the EU/IRIS project in 2003 (Haas, 2003). The aim of the measurements was to test method in the Baltic and obtain data on sea ice ridges in the Baltic. Due to limited flight capacity of the helicopter used, flights were restricted close to coastal region. The 2004 measurements were conducted from Hailuoto island (Haas, 2004b). Aim of the measurement was to collect data on sea ice ridges for developing satellite retrieval methods and numerical models.

HEM measurements in the winter 2005 were part of the pre-launch campaign of the CryoSat satellite mission (Haas & Hendricks, 2005). Based on the satellite images, the flight tracks were designed to capture regions of the most deformed and thickest ice in the Bay of Bothnia in that year. Due to this arrangement, it is expected that the measurements are biased to thick ice and basin wide mean ice thickness is less than the mean ice thickness of HEM measurements.

POL-ICE project in 2005-2007 aimed to determine how operational sea ice monitoring in Finland can best benefit from forthcoming dual-polarized RADARSAT-2 SAR images. The goal of the POL-ICE field campaign in 2007 was to collect sea ice data for development and validation of sea ice products, e.g. ice types, ice thickness, based on dual-polarized ENVISAT and RADARSAT-2. Alfred Wegener Institute conducted helicopter-borne ice thickness measurements with their EM induction sensor over the Bay of Bothnia on 11-14 March (Hendricks et al., 2007). The EM measurements were co-incident as much as possible with the ENVISAT ASAR acquisitions.

Aim of the 2011 campaign was to examine compression of pack ice and to obtain synoptic view of the basin wide ice thickness distribution in the Bay of Bothnia as a part of the SafeWin project (Kujala and Montewka, 2018). The winter 2011 was severe and before the campaign the Bay of Bothnia was covered by thick deformed ice. During the campaign, ice motion was very small and hence the pack ice remained rather unchanged between the daily measurements. Also weather conditions favored the campaign and as an outcome, winter 2011 measurements are the most extensive HEM measurements in the Baltic. The 2011 campaign was used by Gegiuc et al. (2018) as validation data for the ice chart deformation numeral (degree of ice ridging). The agreement between the two datasets was generally good.

## 2.2.2 Ship EM data

EM ice thickness measurements have been made on research cruises in 2012 and 2016 (Table 1). In these, the EM-31 instrument is placed in an enclosure and hung from a boom or a crane outside the ship hull (Haas, 1998). The ship measurements are often biased, as ship crews tend to avoid thickest ice and turn back from impenetrable areas. These effects were minimized by deliberately searching for challenges (2012) or instructing the ship crew to make transects in straight lines whenever possible (2016).

The EM-31 instrument was generally 1-2 meters above sea level during the measurements to avoid impacts from sea ice. Distance to snow surface was measured using an attached laser rangefinder with a 10 Hz measuring frequency and a negligible footprint. Sea ice thickness is calculated separately for each rangefinder reading and these thicknesses are averaged once per meter.

In 2012, ship EM measurements were conducted on board S.A. Agulhas II polar supply and research vessel during maneuver tests in an ice field. The objective of these tests was to test the ship performance in level ice, in ship channels with broken ice, and through pressure ridges in an ice field. Throughout these tests, vibration and forces on the ship hull were measured. The EM ice thickness data was collected both during the maneuver tests and transit periods between them. During the test cruise the ice cover was small and thick ridged ice was only found near the Hailuoto area.

Ship EM measurements from 2016 were part of the Aranda sea ice cruise. The objective of the mission was to carry out a cross-scientific study of the sea ice in the Bay of Bothnia. This included several experiments both during the transit and at the ice stations. Experiments ranged from remote sensing studies to ship transit in ice, basic measurements of sea ice physics and biology in both sea ice and water. The EM data in this study was collected during the transit periods. The cruise track was not set before the mission, but approximate location of an ice station was always decided the night before based on the changing ice conditions. However, as much of the work as possible was planned to take place within the coverage of Hailuoto coastal radar.

## 2.3 Drilling observations

The in-situ ice thickness measurements were made by drilling in the regular monitoring stations in the coastal fast ice zone. The original observations have been made weekly throughout the winter. The length of the time series vary a lot; the longest time series is from Kemi station, where observations have been made since 1912. There is no detailed documentation of the measurement sites available, so the sites might have moved or the environment changed during the longer observation periods. For the period 2003-2016, which is used in our analysis, the Hailuoto station has been located on the southwest coast of the island (64°56´ N, 24°40´ E, see Fig. 1).

**2.4 Atmospheric variables**

To investigate the reasons behind the variability of ice thickness we calculated freezing degree days from air temperature observations in Hailuoto Keskikylä station (65°1´ N, 24°43´ E). Daily mean air temperatures are available since 1959. Freezing degree days means annual cumulative sum of daily mean air temperatures below 0° C.

Another factor affecting the variability of ice thickness is wind. We used days with wind over 14 m/s during the winter months January, February and March (JFM) in Hailuoto Marjaniemi station (65°2´ N, 24°33´ E). The wind has been defined as 10-minute average wind speed and observations have been made 8 times per day since 1984. In some days in the 1980s and in the beginning of the 1990s less observations were made.

In addition, to find out the influence of large-scale atmospheric forcing we used NAO index values from the NOAA Climate Prediction Center (NOAA Climate Prediction Center, 2018). The monthly values were averaged over December, January and February (DJF).

**2.5 Severity of the EM data winters**

The severity of the winters from which we have EM data is presented in Table 2. This does not fully represent the severity of the winters in time period 2003-2016 due to lack of measurements in the two really mild ice winters 2008 and 2015. The years that are presented here include only one winter which is classified as mild. However, five of the seven winters are milder than average between 1961-2010. The sum of freezing degree days is remarkably higher in the two severe winters 2003 and 2011.

**3 Results**

**3.1 Ice charts**

We calculated the results from ice charts over 14 seasons 2003-2016. The length of ice season (Fig. 1) varies from 69 days in the Quark to 157 days in the northern inlets. Basin mean value is 106 days. Self-evidently, ice season is longer in the north than in the south, but ice season is also longer in shallow coastal areas than in the middle of the basin.

The average ice thickness (Fig. 2) does not include days of open water. The values range from 0.11 m by southwest coast to 0.44 m of the northeast land fast ice, the basin average being 0.28 m. The effect of recurring coastal leads can be seen, more prominent along the Swedish coast due to prevailing north-westerly winds. Also the extent of fast ice becomes delineated by the coastal leads. Impact of the prevailing wind pattern is also manifested as enhanced ridging in the north-east sector of the Bay of Bothnia (Fig. 3). Based on ice charts, ridges are common in that regions every year and as an average, duration of ridged or heavily ridged ice conditions is up to 57 days.

In Fig. 4 the seasonal development of the Bay of Bothnia ice area, which has been divided into ice types, is shown. The types are fast ice and the six classes of ice deformation. The values for a given day are averages over the 14 seasons. The area of fast ice expands on average until the mid-March, begins to decline rapidly in mid-April, and disappears at mid-May. Drift ice

disappears after fast ice has melted and that is a feature typical to the basin. In the drift ice the level ice area increases to the beginning of February at which stage also about half of the pack is rafted or ridged. The level and rafted ice areas then decrease gradually while the ridged ice types increase so that in March 75% of pack ice area is rafted or ridged. The total ice area remains quasi constant during February and March before the onset of melting season in April. The apparent faster decline of ridged ice in comparison with level ice in late May is probably in part due to the disappearance of surface ridge signatures in

radar images of the melting pack.

The thickness time series in Fig. 5 is derived for the daily basin averages for fast ice and drift ice thicknesses. Both the scattered values for individual seasons and the average over all 14 seasons are shown. The fast ice growth rate slows down until the thickness reaches 0.52 m before the last third of March. After this the thickness remains constant a month and declines then

rapidly. The average of ice chart ice thickness in drift ice zone, which refers to regionally representative level or slightly rafted types, reaches 0.43 m before the melting period. The pack ice remains about ten days after fast ice has disappeared, probably due to its larger mean thickness.

### 3.2 Helicopter EM data

Grid-averaged HEM data from all years are compared with ice chart data in Fig. 6. The EM data were gathered during several

days before and after the dates of the ice charts. The values in ice chart data are up to 0.5 m. The helicopter EM values exceed the scale, which is up to 1 m, in many measurement points. This is due to the fact that the ice chart values are for ice with flat surface, which is level or rafted ice. The helicopter EM measurements show the real situation with ridged ice in the drift ice zone.

Although the maps are not from exactly the same day from year to year, there is a large inter-annual variability seen in these five years. The mildest ice winter based on both datasets is 2005. In the severest years 2003 and 2011 the entire Bay of Bothnia was ice covered and over 100 cm thick ice was measured also in the south near the Quark.

In ice charts spatially the most severe ice conditions are in the north. In addition, especially in 2003 and 2011 there is more ice

in the eastern side of the bay. The EM data from 2003 and 2005 show that the ice is thicker in the northeastern part. This is because the southwest winds are dominant in the area.

Histograms of grid-averaged HEM ice thicknesses from all measurement points are shown in Fig. 7. The mean of ice thickness in 2003 is 0.98 m and the mode class is 0.6-0.7 m. The data from 2004 (Fig. 7b) are not as evenly distributed as the 2003 data. There is a strong mode of 0.1-0.2 m, representing thin new ice which has grown in coastal polynyas. However, the mean is 1.17 m and thicknesses over 1 m are more common than in other years. The thicker ice indicates that there has been widely

ridged ice. The 2004 campaign was done in the northern Bay of Bothnia in a quite small area near Hailuoto.

The ice thickness from helicopter EM data from 2005 is presented in Fig. 7c. The mean is 0.78 m and the mode class 0.3-0.4 m. The data from 2007 (Fig. 7d) has a high peak. The mean (0.76 m) is close to the mode class (0.5-0.6 m). Thick ice is rare in this campaign, which covered a wide region of the northern Bay of Bothnia. Thus, in 2007 ridged ice was rare. Instead, the

observed distribution points to the fact that level ice and rafted ice might have been the dominating ice types. In 2011 the mean is 0.89 m and the mode class 0.3-0.4 m (Fig. 7e). The average from all five years is presented in Fig. 7f. The mean is 0.92 m and the mode class 0.4-0.5 m.

### 3.3 Coastal boundary zone

EM measured ice thicknesses, ice chart minimum and maximum thicknesses and thickness measurements of fast ice for the

years 2007 and 2011 are combined in Fig. 8. The thicknesses are presented as a function of distance from the fast ice edge. The years were chosen to be the two with the most representative spatial coverage of the coastal boundary zone.

Fig. 8 shows that the EM observed thickness in drift ice zone is thicker than the in situ observed fast ice thickness. The fast ice thickness was 0.51 m (2007) and 0.65 m (2011), whereas mean ice thickness in the drift ice zone closest to the fast ice (< 5

km) was 0.96 m (2007) and 1.21 m (2011).

Effect of the coast on drift ice thicknesses can be seen up to 20 km from the fast ice edge. EM measured thicknesses are consistently larger close to the fast ice edge than further from it.

The variation in EM observed thickness (vertical red bars in Fig. 8) is larger close to the coast than further away from it. In other words, thickest and thinnest ice are found in the zone closer than 20 km from the fast ice edge. The large variability close to the fast ice edge can be explained by open or refrozen coastal leads (thin ice) and fast ice edge acting as a barrier for the ice drift, creating favorable conditions for ridging and accumulation (thick ice).

Fig. 8 shows a large discrepancy between EM observed ice thickness and the ice chart thickness. Based on this, we emphasize that ice chart thicknesses reflect the thickness of undeformed, thermodynamically grown level ice and should not be used as a proxy for overall ice thickness, or volume, of drift ice.

## 3.4 Fractions of deformed and level ice

We calculated the fractions of deformed and level ice from helicopter EM data. As a threshold between the deformed and level ice we used the drilling measurement in Hailuoto station in each year approximately on the 1 March (measurements made weekly). Deformed ice forms later than the fast ice and therefore is younger and level ice thickness in the drift ice zone is not able to grow thicker than fast ice. That is why we use the fast ice thickness measurement as a threshold. By deformed ice we mean here rafted and ridged ice and rubble field.

Table 3 shows the fractions of deformed and level ice both in the entire measurement area and in the area west of Hailuoto for all of the helicopter EM data available. The area west of Hailuoto has been chosen because there are measurements from all years in the area. The limits of the area are in latitude 64.5° N-65.5° N and in longitude 23° E-24.5° E (see Fig. 6).

For all of the campaigns, areal fraction of deformed ice is considerable. The minimum fraction for deformed ice of 49.7% was found during 2011 and the maximum of 70.4% during 2007. Interestingly, during the two severe ice winters (2003 and 2011) in our dataset we find the smallest (2011) and third smallest (2003) areal fractions of deformed ice.

Area west of Hailuoto is usually an area where is much deformation. The fraction of deformed ice in 2003-2007 was 60-70%. Interestingly, the fraction was only 43.3% in 2011. Recalling that the winter 2011 was severe in the Baltic and consequently thermodynamically grown ice was thick enough to reduce ice motion. That might explain the smaller fractions of deformed ice in severe winters in the entire Bay of Bothnia data, too.

## 3.5 Inter-annual variability

In Fig. 9 we have compared the different ice thickness data sets to each other in winters between 2003 and 2016. Drilling observations from Hailuoto represent the fast ice zone. The ice chart data, helicopter EM mean data and ship EM mean data are averages over drift ice area west of Hailuoto (see Fig. 6). Helicopter EM mode and ship EM mode are modes over the same drift ice area west of Hailuoto. In this figure the EM mean and mode values have been calculated from the entire data differently from the histograms. The ice charts are from 1 March and open water has not been taken into account in mean values. The drilling measurements are made weekly, so the day of the measurements varies between 26 February and 4 March. Figure 8 shows also the NAO index values for each winter. In addition, we have listed the amount of freezing degree days, wind days and the value of NAO index in Table 4 together with the ice thickness values.

The highest values of ice thickness are in the helicopter EM data, which indicates the drift ice thickness. The histograms of the helicopter EM data in this area are in Fig. 10. To avoid areal focus we have also calculated histograms in 1 NM grid for each

grid point and then averaged all histograms from the grid points. The mean and mode marked in histograms are calculated from the histograms unlike the values in Fig. 9 and Table 4.

In 2004, when all the measurements were made in this area, there was a lot of thin ice. Also thick, ridged ice exists during this average winter in terms of freezing degree days. These kind of ice conditions would be expected when there is a lot of wind. However, in 2004 there has been only 10 days of wind over 14 m/s, which is less than average of 2003-2016.

Compared to the results of the entire campaign in 2005 there is more thick ice in the selected area. This is clearly seen also in Fig. 6, where the thickest ice is found in northern parts of the campaign in 2005. Year 2007 was mild and there were 9 wind days. Thus, the thick ice is missing almost completely. In 2011 there is a higher peak in the histogram of the selected area compared to the histogram of the entire campaign. The 2011 campaign was the broadest of all campaigns. It was the most severe winter in terms of freezing degree days, but in spite of the 15 winds days there was not much thick, ridged ice.

The histograms of ship EM from the selected area are also included in Fig. 10. In both winters 2012 and 2016 the modal thickness is 0-0.1 m. Especially in 2016 there has been a lot of thin ice and the mean is only 0.29 m. This winter was the mildest from the EM data years in terms of freezing degree days and maximum annual ice extent in the Baltic Sea.

The ice chart mean values show that there is much variability between years in the drift ice zone level ice thickness. Based on ice charts the mildest ice winters have been 2015 and 2008. Instead, in the drilling observations year 2009 was milder. In the freezing degree days mildest winter was 2014 and the severest 2011. The most wind days occurred in the mild winter 2015. Nevertheless, the second windiest winter was the severe winter 2003 where the EM data has its highest values and there was much ridged ice.

## 4 Discussion

Our data shows that large inter-annual variability exists between ice seasons, although the EM data from different years is not directly comparable. In some years areas with heavily ridged ice form and there is much deformed ice in the Bay of Bothnia. However, in other years, for example in 2007, thinner ice dominates and there is no thick ice. Winter 2004 was milder and less windy than average. Yet, we found that both thick, ridged ice and thin, new ice formed during that year. In 2015 the Bay of Bothnia remained partly ice-free for the first time so that it has been reliably recorded by satellites (Uotila et al., 2015). In our atmospheric data the winter 2015 was the windiest and one of the mildest. The ice chart mean ice thickness in the area west of Hailuoto indicates level ice thickness of only 0.12 m. The winter air temperature was still so low that in Hailuoto there was 0.62 m fast ice in 2 March 2015. Generally, ice thickness in the fast ice varies much less than the thickness of drift ice.

The average extent of the fast ice zone can be seen in Fig. 2. In Hailuoto the long-term maximum ice thickness in the fast ice zone in years 2003-2016 was 0.86 m. That was reached in 2010. The maximum value is usually reached in mid-March. The record value is from 1985 in Tornio, 1.22 m (Leppäranta and Myrberg, 2009). The most ridged areas are in the northeastern Bay of Bothnia near the line of the fast ice zone. This is because of dominating winds from southwest. Even in mild winters

high ridges form in the Bay of Bothnia. However, they cover a smaller area than in more severe winters.

Heavily ridged areas are found near the fast ice boundary. In the Bay of Bothnia they lie mostly in the northeastern corner as can be seen from Fig. 3. Oikkonen et al. (2016) found out that the drift of ice was anisotropic on coastal boundary zone and that was due to effect of the coast. The alongshore component of the drift was larger than the cross-shore component. The drift

speed was smaller near the fast ice (Oikkonen et al., 2016), which is consistent with our result that ice is thicker near the fast ice edge. The thicker the ice is, the less it moves. Our EM data shows that the ice thickness in the heavily ridged areas in scales of tens of km$^2$ can be much thicker than the fast ice. These areas are challenging for winter navigation and biological hot spots in spring as the ice melts last. Our results in section 3.3 highlight the importance of coastal boundary zone for ice production. In these areas ridging or opening of leads are constantly occurring.

The effect of large-scale atmospheric circulation on ice extent and ice concentration has been studied in the Baltic Sea (Vihma and Haapala, 2009; Vihma et al., 2014). Positive NAO index values indicate milder ice conditions and negative NAO index values more severe conditions. In addition, Koslowski and Loewe (1994) showed that accumulated areal ice volume was negatively correlated with NAO index in a small area in the southwestern Baltic Sea. Research on the correlation of NAO

purely with ice thickness is still lacking. However, ice thickness is a more complex variable than for example ice extent. Winters with strongly positive NAO index, such as 2015, are generally mild and windy. Wind piles up the ice and conditions can be like in 2004, when in our study both thick, ridged ice and thin, new ice existed.

The correlation coefficient of the NAO index and the ice chart level ice thickness is -0.53 in our study period 2003-2016 (Fig.

11). The correlation of NAO and drilling data is -0.38. Neither of the correlations is statistically significant. Thus, our study does not show any significant correlation between NAO and level ice thickness. However, our time period is too short to capture the long-term behavior of the variables. The shortness of our data sets restrict us to determine the effect of NAO on ice thickness. However, our hypothesis is that in winters with negative NAO more ice is produced. Our study period is too short for defining climate variations, too. Especially because the inter-annual variability in the area is so large.

The amount of days of wind over 14 m/s has no notable correlation with the amount of freezing degree days. This can be due to that circumstances change during the winter, like in 1986 when NAO was strongly negative in February and positive in other winter months (Vihma and Haapala, 2009).

Although studies concerning variability of deformed ice have not been carried out in the Baltic Sea, our results can be compared to studies from Arctic and Antarctic. King et al. (2017) used helicopter EM data from years 2003 and 2014 from the Barents Sea to determine the sea ice thickness. As in our study, large inter-annual and spatial variability was found. In 2003 regional modal ice thicknesses were 0.6-1.4 m and in 2014 0.5-0.8 m (King et al., 2017). When comparing to Arctic and Antarctic

surveys, has to be remembered that there is no multiyear ice in the Baltic Sea. For example, EM measurements in Fram Strait between 2001 and 2012 show decrease in both modal and mean ice thicknesses (Krumpen et al., 2016). However, the age of the ice has decreased from 3 to 2 years in 1990-2012. As also in our study, because of the short time series, definite conclusions of thinning trend in ice thickness cannot be done from the study of Krumpen et al. (2016).

In the Antarctic, Sugimoto et al. (2016) also found large inter-annual variability in sea ice thickness by using ship-borne EM data and visual observations in 2001-2012. From their analysis can as well be seen the same tendency than from our results that the ice thickness is larger near the fast ice edge. Worby et al. (2008) defined that the total Antarctic sea ice thickness from ship-based observations was 0.87 +/- 0.91 m. In our study, the mean of the drift ice thickness is 0.92 m, whereas the standard deviation is remarkably smaller (0.33 m), although our study area is smaller. Worby et al. (2008) also concluded that the mean

ice thickness, including ridges, was 40% greater than the mean level ice thickness. Our results show that the in the Bay of Bothnia the mean ice thickness from helicopter EM measurements is 217% greater than the mean level ice thickness from ice charts.

## 5 Conclusions

We have examined different data sets of ice thickness from the Bay of Bothnia in order to define the inter-annual variability

of sea ice thickness. Different data sets describe different parts or different ice types in the Bay of Bothnia. Nevertheless, we found large variability both in time and space.

The inter-annual variability in the fast ice zone is much smaller than in the drift ice zone. Deformed ice has a major role in the drift ice zone. In some years there is mainly thin, thermodynamically grown ice even in the drift ice zone and in some years

large areas with thick ridges. Most of the years are a mixture of these two. Ice thickness varies even in few days time, especially in the drift ice zone. Thus, our observations do not describe the absolute inter-annual variability.

Also large regional differences can be detected. The driving forces are wind and air temperature. Ice conditions are more severe in the north because of the colder air temperatures. Therefore the ice in this region is older and experiences more deformation

to accumulate more ridges. Typical wind direction in the area is southwest. Southwest winds gather ice towards the northeastern area of the Bay of Bothnia, where ice conditions are more severe. The southwest corner is the mildest.

In the drift ice zone the fraction of deformed ice is over half from total volume. In heavily ridged ice regions near the fast ice edge, mean ice thickness is 0.45-0.56 m thicker than in pure thermodynamically grown fast ice.

We emphasize that the ice thickness indicated in the ice charts underestimate real ice thickness of drift ice. Ice charts are based on limited number of undeformed ice observations, but in particularly, mean ice thickness in the coastal boundary zone can be 2-3 fold compared to the pure thermodynamically grown undeformed ice.

Our attempt to solve the inter-annual variability showed that the winters are really different from each other. However, there is so much variability between years and between measurement methods that the results must not be seen as absolute differences from year to year, only as guidelines for ice thickness changes in the seasonal sea ice zone.

*Data availability:* The helicopter EM data from years 2003, 2004, 2005 and 2007 are available at PANGAEA (https://www.pangaea.de/).

*Competing interests.* The authors declare that they have no conflict of interest.

*Acknowledgements.* This research has been supported by a grant from the Vilho, Yrjö and Kalle Väisälä Foundation. Airborne EM surveys were funded by EU projects IRIS and SafeWin. Andi Pfaffling, Stefan Hendricks, and Alec Casey contributed to the HEM data collection.

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

**Table 1: The EM data year, dates, instrument and notes of the campaign.**

| Year | Dates | Instrument | Notes |
|------|-------|------------|-------|
| 2003 | 20-21 February | Helicopter-towed EM-bird | IRIS |
| 2004 | 5 February - 17 March | Helicopter-towed EM-bird | IRIS |
| 2005 | 10-16 March | Helicopter-towed EM-bird | CryoVEx |
| 2007 | 11-14 March | Helicopter-towed EM-bird | POL-ICE |
| 2011 | 2-7 March | Helicopter-towed EM-bird | SafeWin |
| 2012 | 21-22 March | Ship-borne Geonics EM-31 | SA Agulhas II ice test |
| 2016 | 1-10 March | Ship-borne Geonics EM-31 | Aranda sea ice cruise 2016 |

**Table 2: Maximum annual ice extent from the entire Baltic Sea (MIB), MIB anomaly based on years 1961-2010 and severity according to Finnish Ice Service compared to North Atlantic Oscillation (NAO) index and the amount of freezing degree days in Hailuoto. The data is from the years we have electromagnetic measurements.**

| Year | MIB (km$^2$) | MIB anomaly | MIB severity | NAO index | FDD (degrees) |
|------|------|------|------|------|------|
| 2003 | 233000 | + | Severe | -0.05 | 1345 |
| 2004 | 153000 | - | Average | 0.07 | 879 |
| 2005 | 178000 | - | Average | 0.89 | 773 |
| 2007 | 140000 | - | Average | 0.36 | 822 |
| 2011 | 309000 | + | Severe | -0.67 | 1430 |
| 2012 | 179000 | - | Average | 1.37 | 779 |
| 2016 | 110000 | - | Mild | 1.31 | 670 |

**Table 3: The fraction of undeformed and deformed ice in helicopter EM measurements in the entire Bay of Bothnia and in the area west of Hailuoto. The drilling measurement in Hailuoto station in each year approximately on the 1 March has been used as a threshold between undeformed and deformed ice.**

| Year | Entire Bay of Bothnia | | Area west of Hailuoto | |
|---|---|---|---|---|
| | Undeformed ice | Deformed ice | Undeformed ice | Deformed ice |
| 2003 | 48.4% | 51.6% | 36.9% | 63.1% |
| 2004 | 36.1% | 63.9% | 36.1% | 63.9% |
| 2005 | 50.1% | 49.9% | 38.8% | 61.2% |
| 2007 | 29.6% | 70.4% | 31.1% | 68.9% |
| 2011 | 50.3% | 49.7% | 56.7% | 43.3% |

**Table 4: Ice thickness observations, atmospheric variables and their statistics from winters 2003-2016. Drillings are ice thickness drillings from Hailuoto approximately 1 March, ice charts data is mean from the area west of Hailuoto 1 March (open water not taken into account), EM/mean is the EM mean ice thickness in the same area and EM/mode the mode thickness in the area. In EM data HEM means helicopter-borne EM and SEM ship-borne. FDD is the amount of freezing degree days during the winter in Hailuoto, wind days are days with wind over 14 m/s in JFM in Hailuoto and NAO index is average value for winter months DJF.**

| Year | Drillings (m) | Ice charts (m) | EM/mean (m) | EM/mode (m) | FDD (degrees) | Wind days (d) | NAO index |
|------|------|------|------|------|------|------|------|
| 2003 | 0.81 | 0.41 | 1.42 (HEM) | 0.41 (HEM) | 1345 | 18 | -0.05 |
| 2004 | 0.55 | 0.25 | 1.18 (HEM) | 0.17 (HEM) | 879 | 10 | 0.07 |
| 2005 | 0.55 | 0.19 | 1.06 (HEM) | 0.37 (HEM) | 773 | 15 | 0.89 |
| 2006 | 0.42 | 0.28 | | | 1090 | 8 | 0.10 |
| 2007 | 0.51 | 0.24 | 0.75 (HEM) | 0.57 (HEM) | 822 | 9 | 0.36 |
| 2008 | 0.46 | 0.13 | | | 521 | 7 | 0.66 |
| 2009 | 0.39 | 0.26 | | | 728 | 13 | -0.08 |
| 2010 | 0.82 | 0.36 | | | 1293 | 6 | -1.67 |
| 2011 | 0.65 | 0.45 | 0.84 (HEM) | 0.40 (HEM) | 1430 | 15 | -0.67 |
| 2012 | 0.48 | 0.34 | 0.77 (SEM) | 0.57 (SEM) | 779 | 11 | 1.37 |
| 2013 | 0.65 | 0.46 | | | 1183 | 8 | 0.02 |
| 2014 | 0.57 | 0.31 | | | 509 | 15 | 0.86 |
| 2015 | 0.62 | 0.12 | | | 512 | 24 | 1.66 |
| 2016 | 0.67 | 0.30 | 0.42 (SEM) | 0.07 (SEM) | 670 | 5 | 1.31 |
| Mean | 0.58 | 0.29 | 0.92 | 0.37 | 895 | 12 | 0.34 |
| St. Dev. | 0.13 | 0.11 | 0.33 | 0.19 | 318 | 5 | 0.88 |
| Median | 0.56 | 0.29 | 0.84 | 0.40 | 801 | 11 | 0.23 |
| Min | 0.39 | 0.12 | 0.42 | 0.07 | 509 | 5 | -1.67 |
| Max | 0.82 | 0.46 | 1.42 | 0.57 | 1430 | 24 | 1.66 |

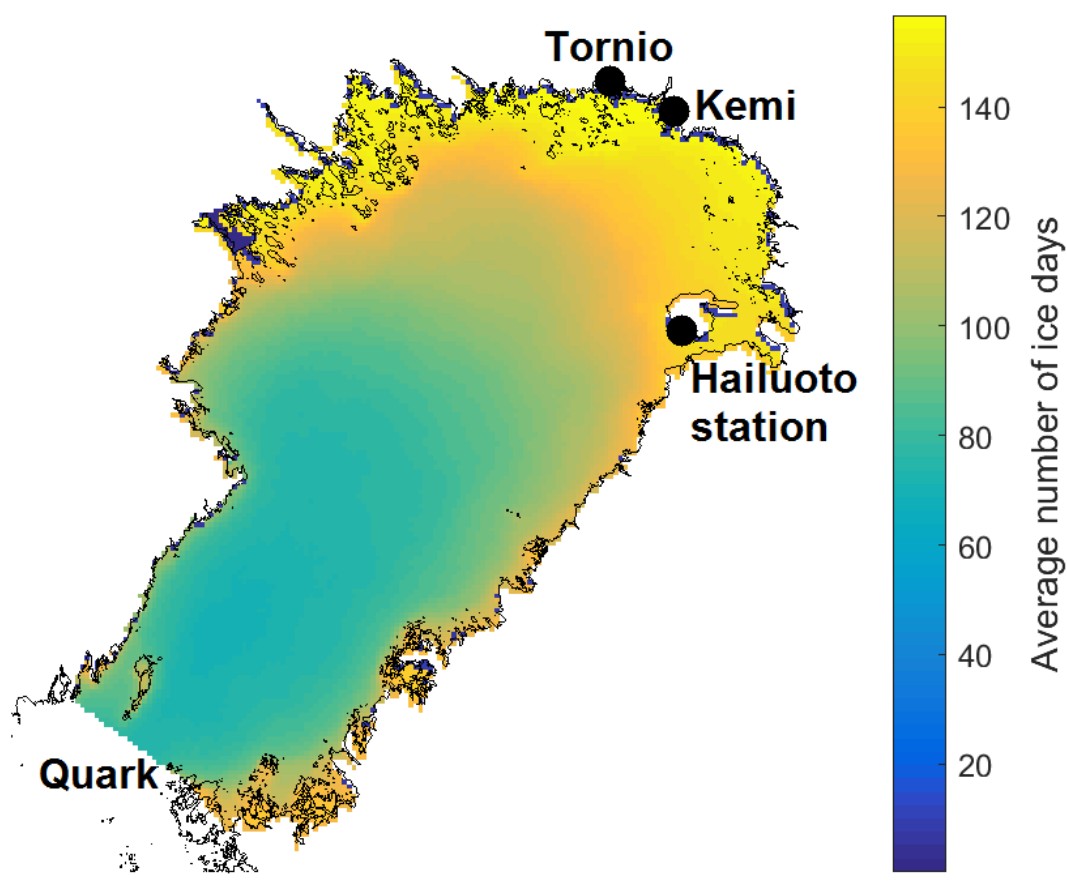

**Figure 1: The average number of ice days per season determined from ice charts and the location of places mentioned in text. Hailuoto station is the location of the ice drilling station at the Hailuoto island.**

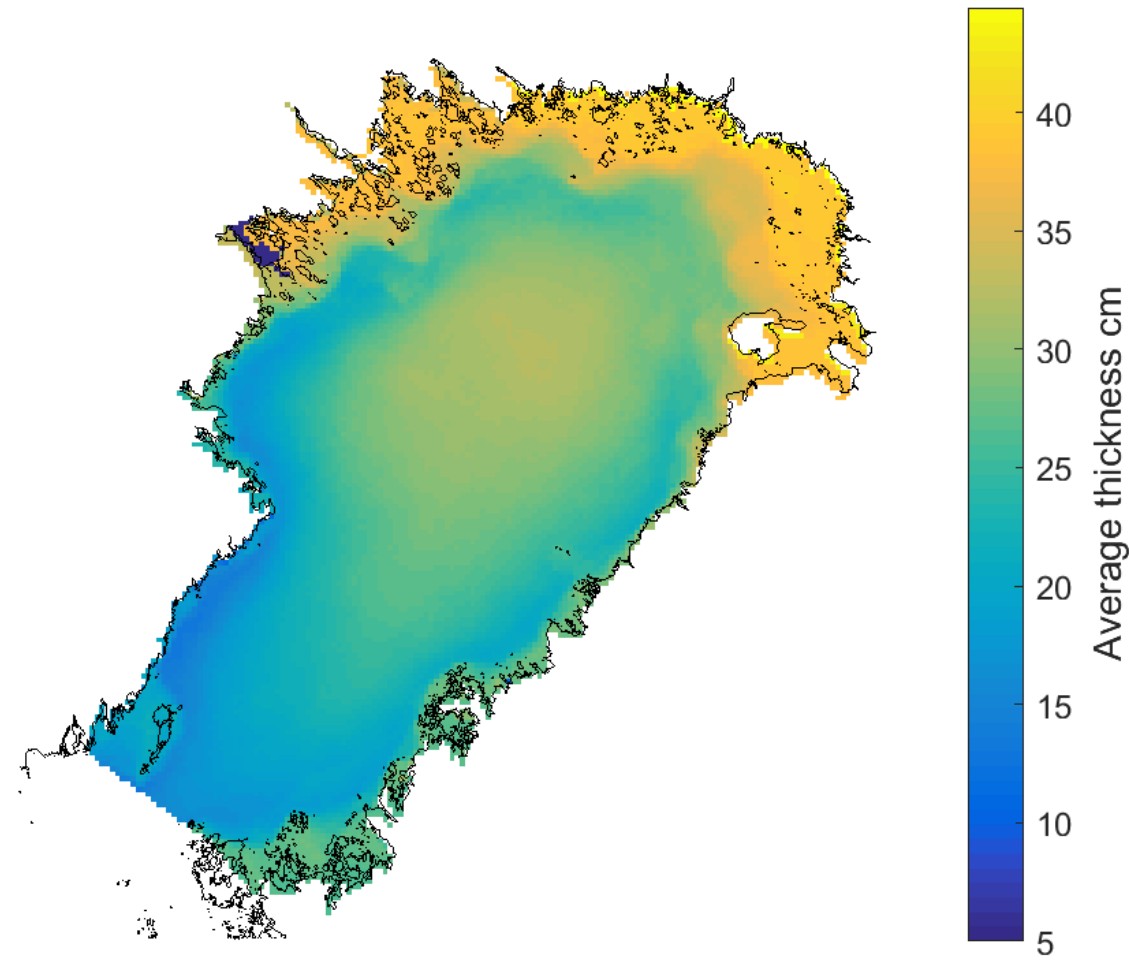

**Figure 2: The average seasonal, level ice thickness from ice chart data. Values are averages of nonzero thickness values over 14 seasons.**

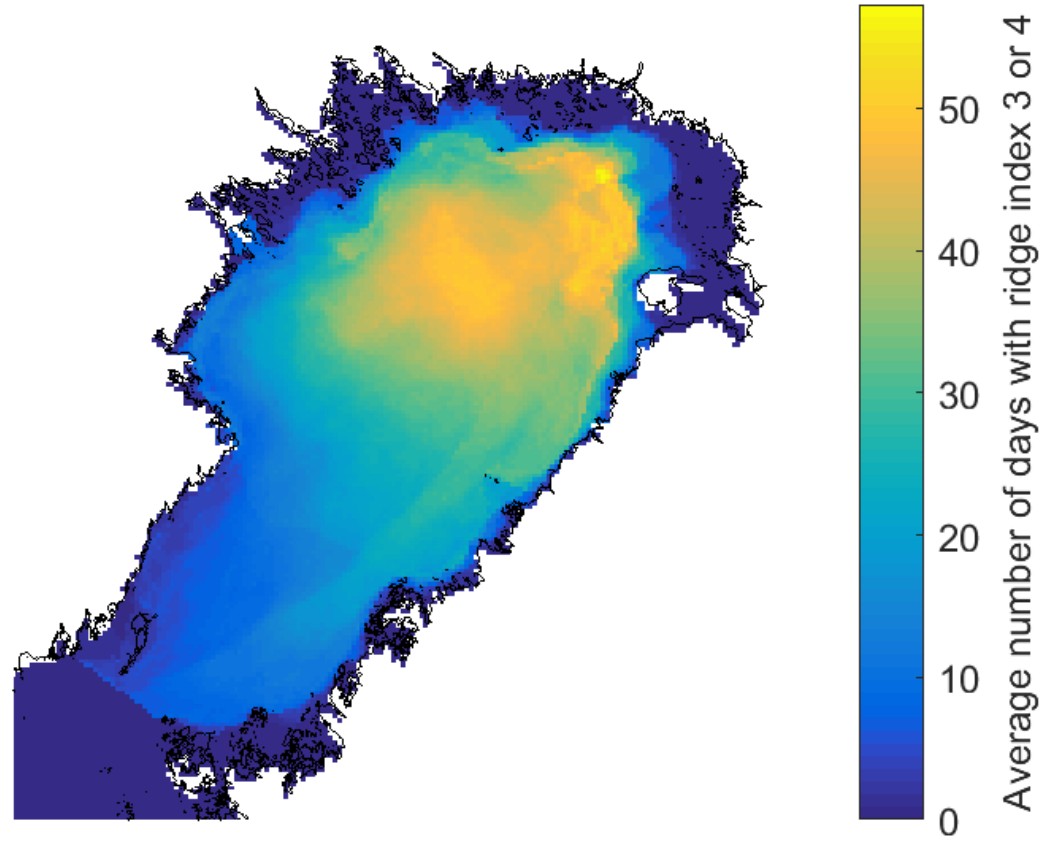

**Figure 3: Average number of days with ice chart ridging index 3 (ridged ice) or 4 (heavily ridged ice).**

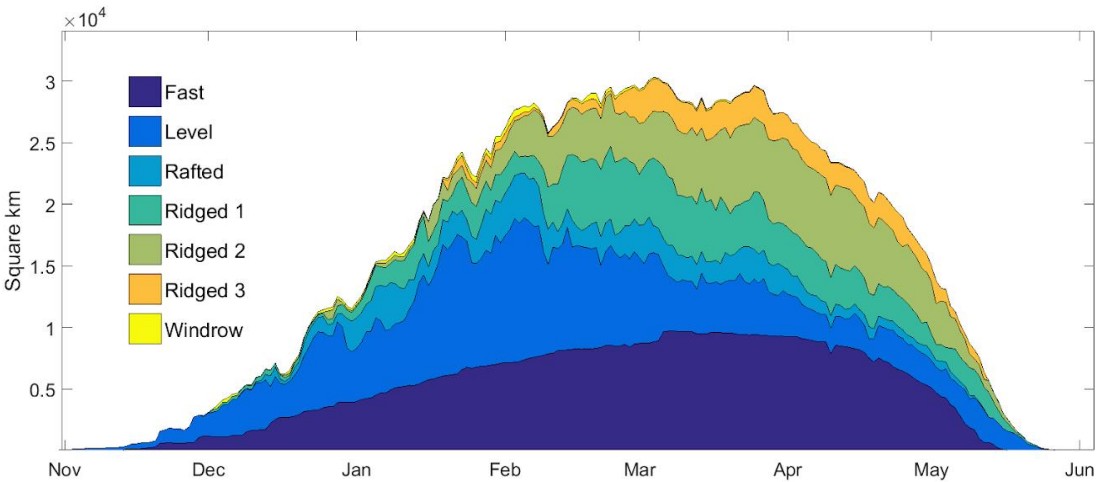

**Figure 4: The seasonal development of charted ice area as divided into ice types. Season day averages over 14 seasons, including open water.**

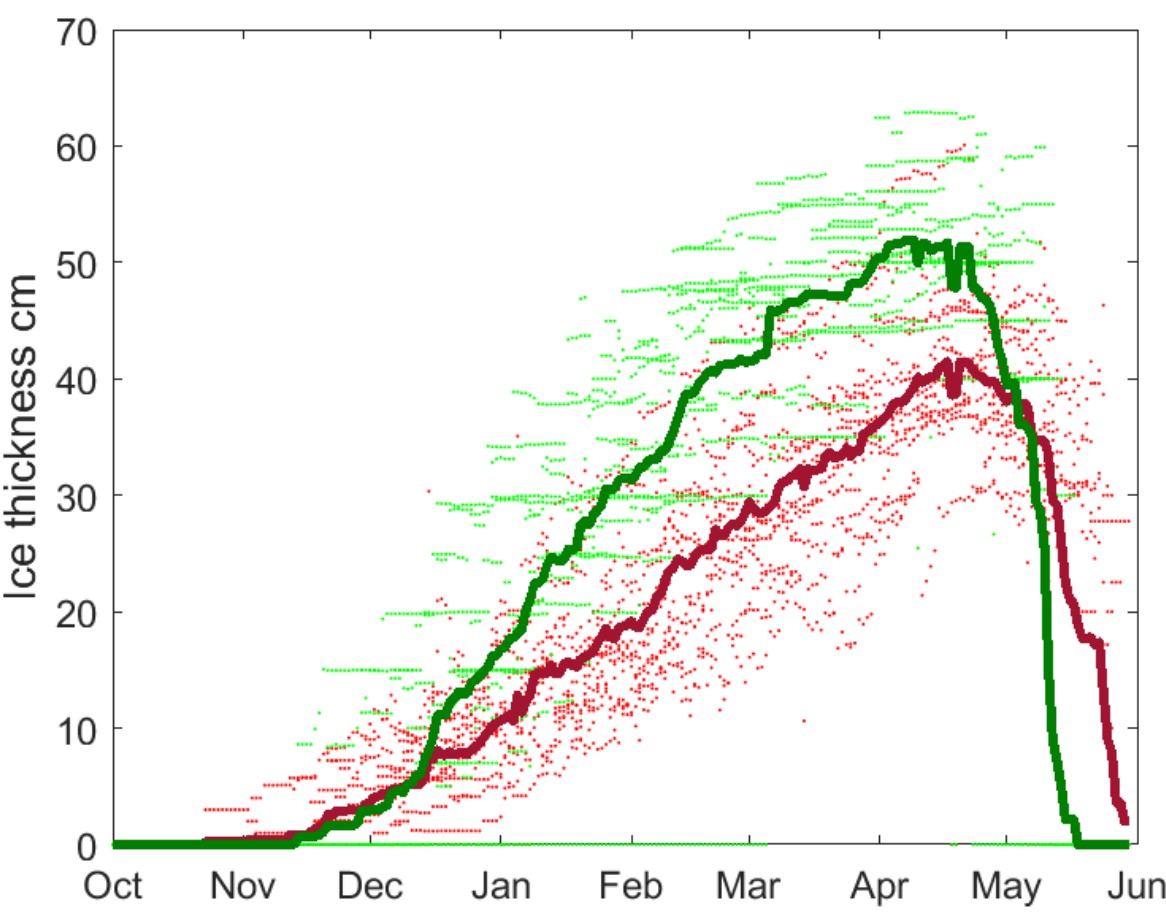

**Figure 5: Seasonal development of level ice thickness. Data is based on ice chart information thickness for 14 seasons 2003-2016. The green and red dots are daily level ice thicknesses in the fast ice and pack ice zones for different seasons, the green line is fast ice thickness average over seasons and red line pack ice thickness.**

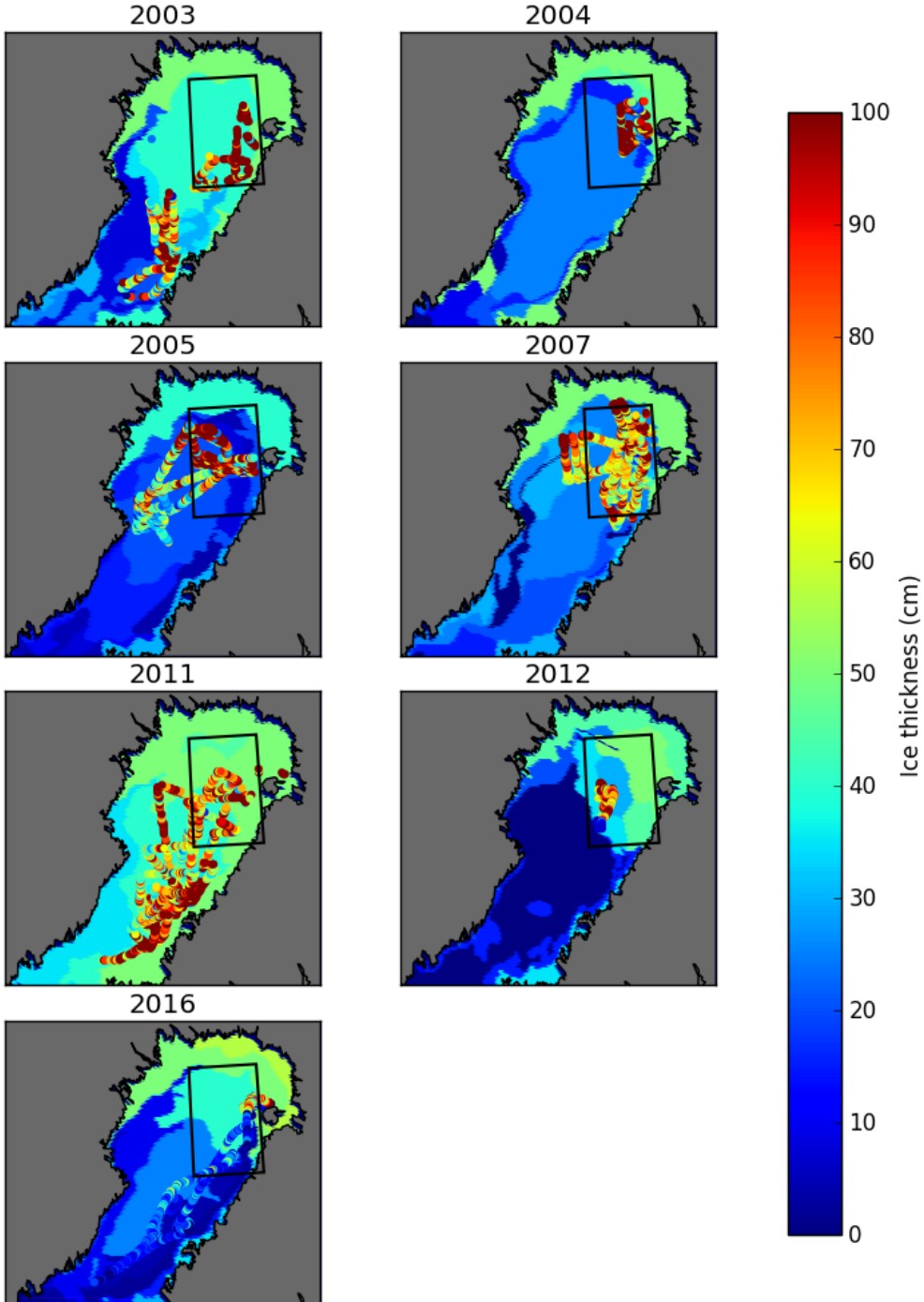

**Figure 6: Ice chart (on the background) and 1 nautical mile averaged EM data (with dots). Ice charts are from 23 February 2003, 14 March 2004, 13 March 2005, 14 March 2007, 4 March 2011, 21 March 2012 and 5 March 2016. The dates of EM data are in Table 1. As we can see, the ice charts report uniform, thin ice, while in reality thick deformed ice types are typical. The black box is the area west of Hailuoto used in analysis. The limits of the area are in latitude 64.5° N-65.5° N and in longitude 23° E-24.5° E.**

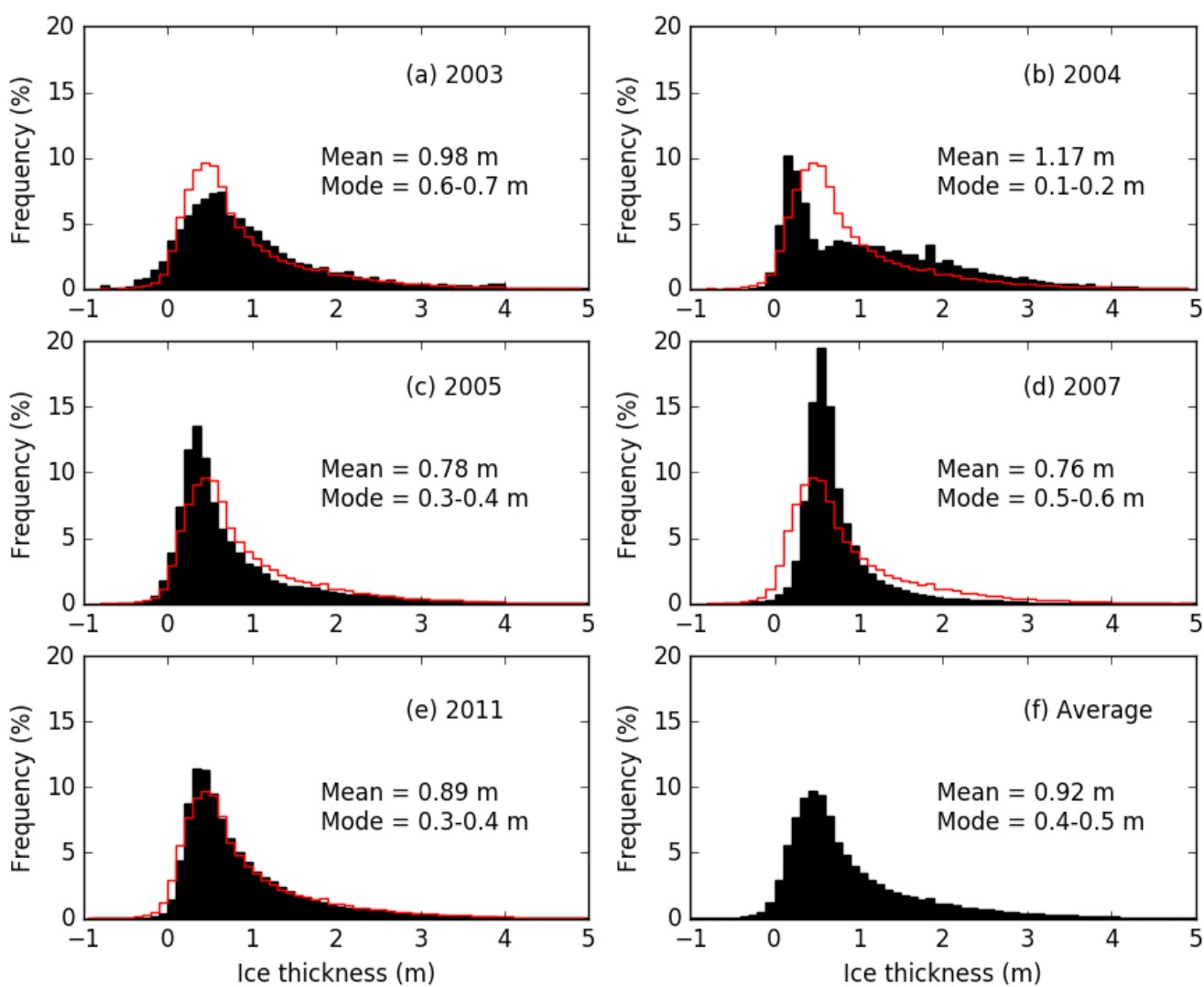

**Figure 7: Frequency histograms of all helicopter EM ice thickness measurements in a) 2003, b) 2004, c) 2005, d) 2007 and e) 2011 (in black). The average of all years is in figure f and in red in other figures. To avoid areal focus we have first calculated histograms in 1 NM grid for each grid point and then averaged all histograms from the grid points.**

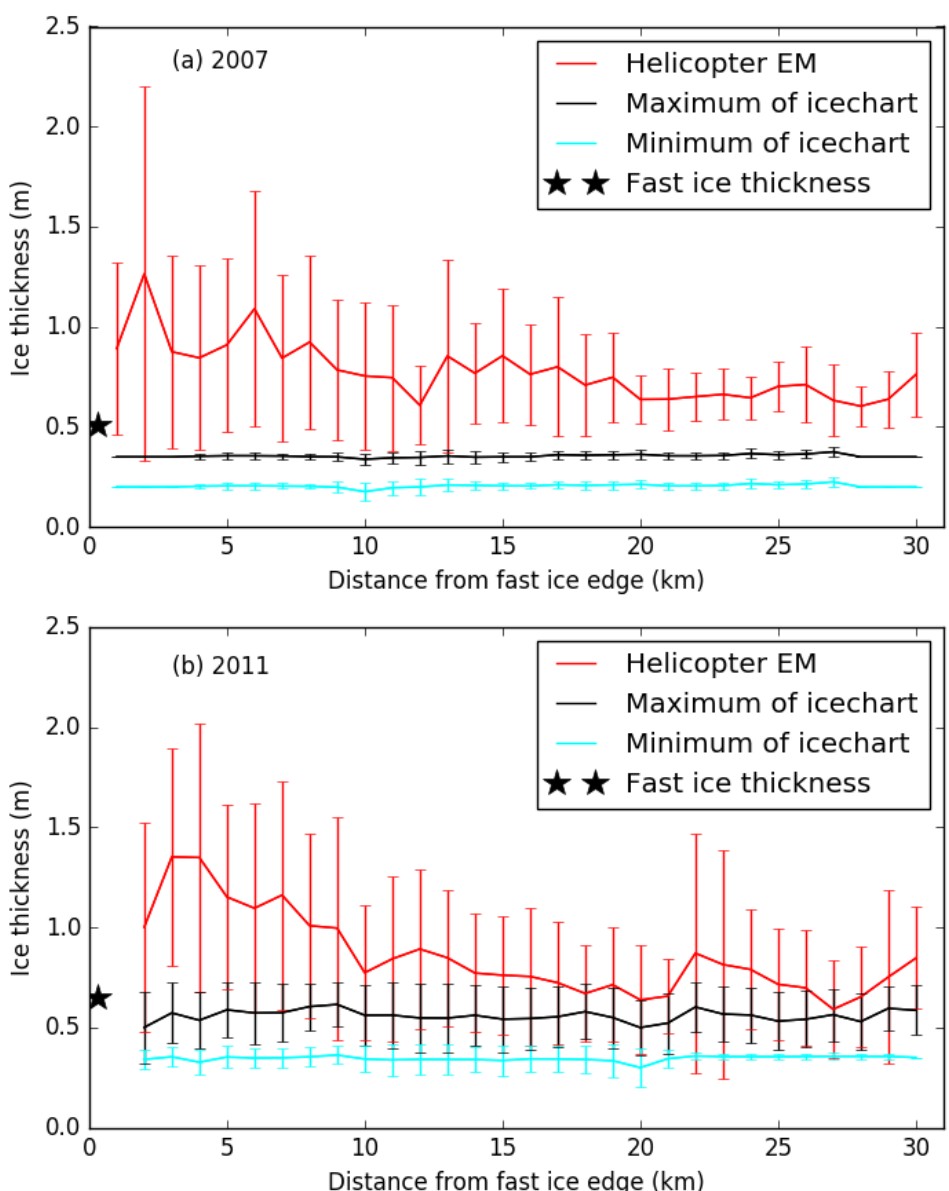

**Figure 8: Ice thickness a) 2007 and b) 2011 from helicopter EM (red) and maximum (black) and minimum (cyan) of ice charts from the same points where helicopter EM was measured. Ice thickness is shown with the distance from fast ice edge. The bars are ± standard deviation and the stars indicate fast ice thickness on approximately 1 March in Hailuoto station.**

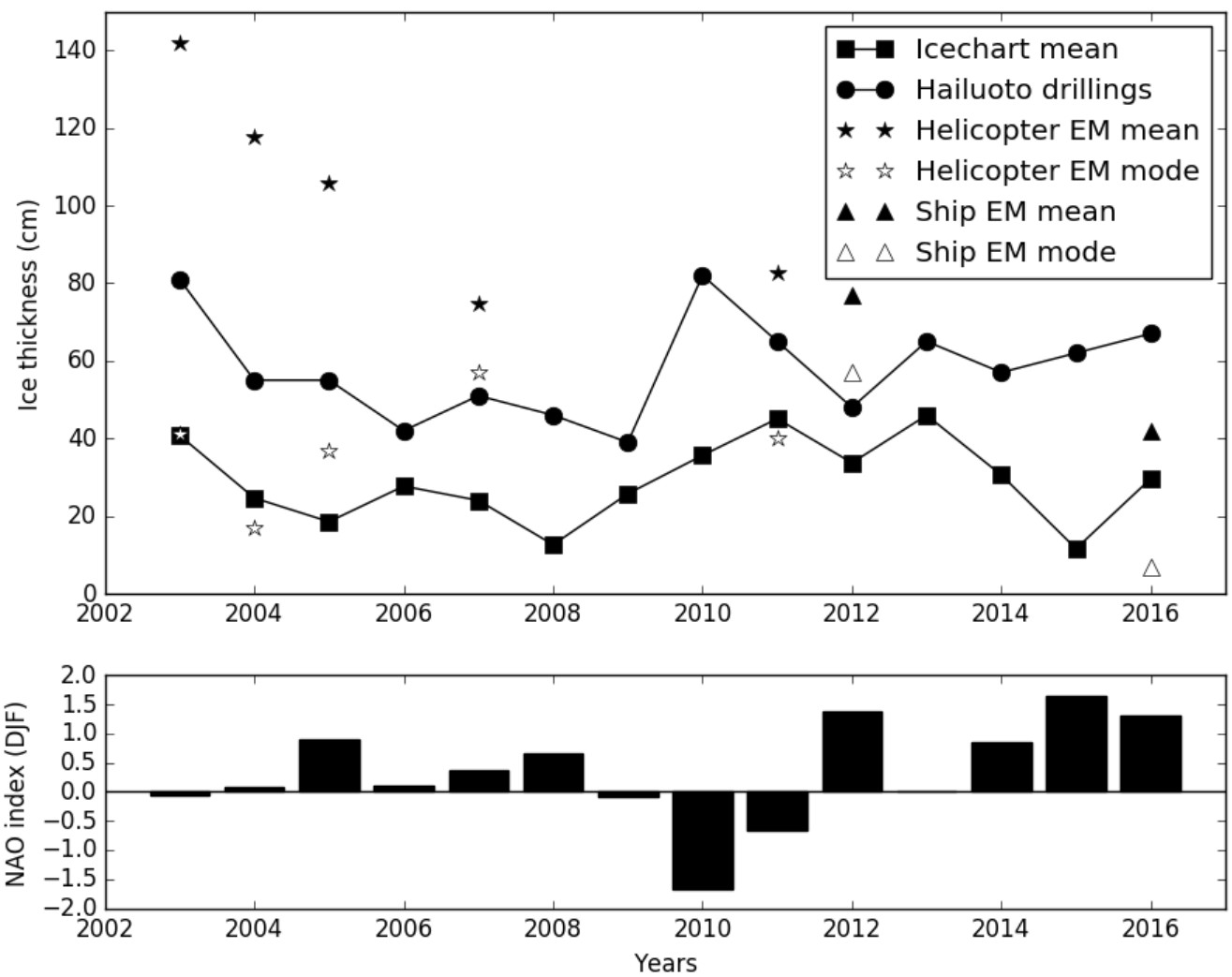

**Figure 9: Time series of mean and modal ice thicknesses from Hailuoto drillings (circles) in the fast ice zone, and ice charts (squares), helicopter EM surveys (black stars) and ship EM surveys (black triangles) in the drift ice zone west of Hailuoto (upper panel). Helicopter EM modes (white stars) and ship EM modes (white triangles) were also observed over drift ice west of Hailuoto. Lower panel shows time series of mean NAO index of winter months DJF. Data from years 2003-2016.**

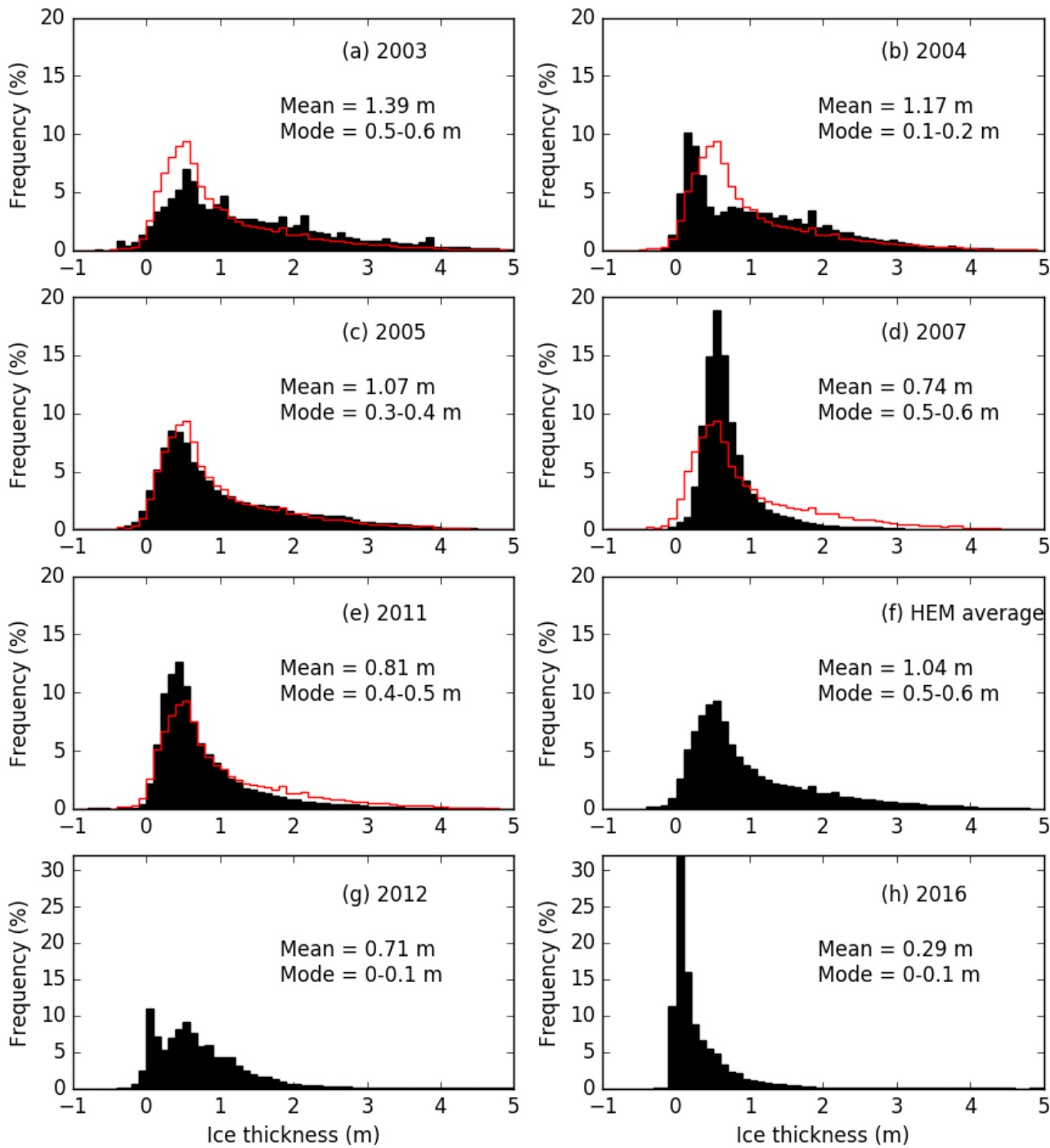

**Figure 10:** Frequency histograms of helicopter EM ice thickness in the drift ice area west of Hailuoto a) 2003, b) 2004, c) 2005, d) 2007 and e) 2011 (in black). The average of all years of HEM is in figure f and in red in other figures. Frequency histograms of ship EM ice thickness in the drift ice area of of Hailuoto g) 2012 and h) 2016. To avoid areal focus we have first calculated histograms in 1 NM grid for each grid point and then averaged all histograms from the grid points.

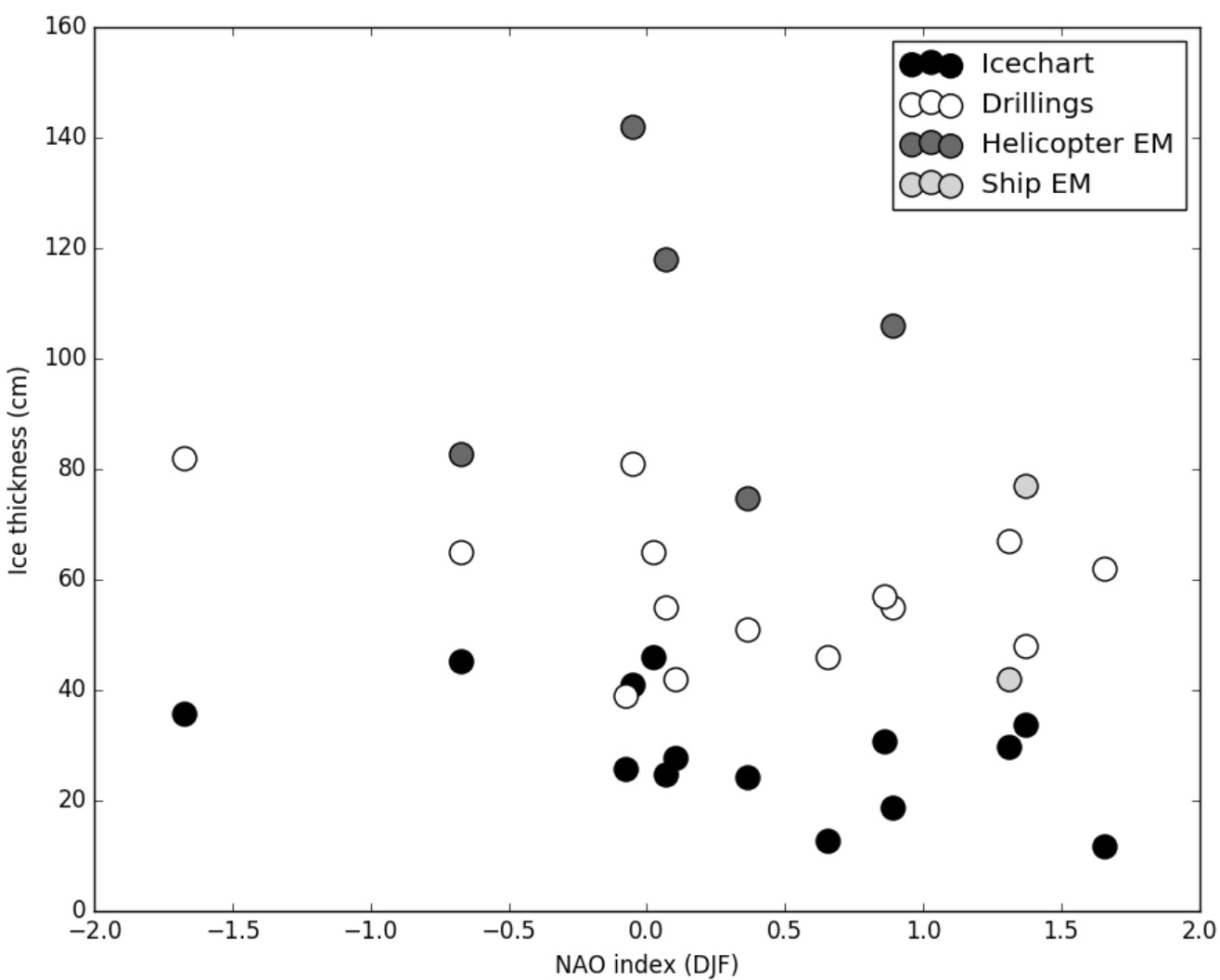

**Figure 11: NAO index (DJF) and ice thickness from ice charts (black), drillings (white), helicopter EM surveys (dark grey) and ship EM surveys (light grey) 2003-2016.**

