# Peer review of "Inter-annual sea ice thickness variability in the Bay of Bothnia"

_The Cryosphere, 2018_

## Referee Comment (RC1) · Anonymous Referee #1 · 12 Jun 2018

The paper presents ice thickness data from EM measurements and drillings in the framework of level ice thickness indications in Baltic ice charts. The data are new and valuable and deserves publication.

However, the paper needs to be cleaned up. The language is often unclear (or poor English) and important information for the reader less familiar with the Bay of Bothnia is missing or hard to find.

There is a lack of discussion of uncertainty/error sources such as the sampling strategy. Not the same areas are sampled by the HEM each year but this is hardly discussed in the paper. This should feature more prominently in the discussion/conclusion sections.

Specific comments:

P1L14: in a pure -> in pure P1L16-30: A map with the place names that are referred to in the paper would be very helpful. Perhaps figure 1 could be used. P2L2 and L15. These statements say almost the same but still are slightly contradictory. Does all ice grow thermodynamically in the fast ice zone, or is this only the case in more sheltered areas. A follow up question is if the areas where the drillings and EM measurements are carried out can be considered 'more sheltered'? P3L4: What does 'the entire time series' men. Please give start and end years. P3L8: This statement seems to be contradicted by the conclusions section. Perhaps it is more the interannual variability of ice extent that is related to NAO, and not the thickness. The term 'ice conditions' normally refers to ice extent AND thickness etc. I suggest to change ice conditions to ice extent here. P3L30: It depends … (what depends?) Paragraph should be rewritten. P3L31: We are now using … (since when?) Do you mean 'In this study we use …'? P3L32-33: from fast ice zone -> from the fast ice zone P4L1: Here I guess by conditions you mean thickness? P4L15: Refer to figures 1-3 for an outline of the area of interest. P4L26: satellite images and observations. Which observations?? P4L28: upturning -> turning P5L5: of degree -> of a degree P6L4: of approximately for HEM bird -> ?? Section 2.2, 2.2.1 and 2.2.2: There is some redundancy in the 3 EM sections. Perhaps these could be combined. P7L20: solve -> investigate P7L29: have been -> were P8L5: not represent -> not fully represent P8L2-5: Note that there is a cyclic feedback between FDD and ice extent (and thickness) in the sense that lots of FDDs leads to more and thicker ice, but also that lots of ice shelters the warm ocean from the atmosphere and thus leads to more FDDs. P8L8: The results … (results of what??, this is a bad way to start a sentence) P8L23: three fourths -> use percentage instead (e.g. 75% although from figure 4 it seems to be less than 75%) P9L1: The growth before this is linear. This may be true in the average sense, but most likely not in the individual years, so please be more specific. P9L11: data -> datasets P9L18: Histograms of … A specification of the area of interest is needed here. (and perhaps a discussion of the sampling of this area each of the years) P9L18 (and fig 7): I notice negative ice thicknesses in the histograms. This needs a comment/explanation.

P10L1: drift ice area west of hailuoto – show on a map (f.ex. figure 6) P10L12: have been -> were P10L13: there is a lot -> there was a lot P10L17: That is clearly -> This is clearly P10L30-31: I suggest to rephrase to: severe winter 2003 where the EM data has its highest values and there was much (or a large fraction of) ridged ice. P11L5-6: for the first time so that it has been reliably recorded ??? P11L6: that winter -> the 2015 winter P11L11-12: maximum ice thickness in the fast ice zone in years 2003-2016 was over 0.8. So the max in each year was above 0.8m or in just one of the 14 years?? P12L17-18: In a similar way …. - Please rewrite sentence to make more sense. P12L23: but the reality is much more severe – Please rewrite. The level ice thickness given in the charts is also a part of the reality.

Note that the above specific language corrections are not the only ones needed for the paper to become acceptable.

---

## Referee Comment (RC2) · Anonymous Referee #2 · 29 Jun 2018

General Comments: This paper studies the evolution of ice thickness in the Bay of Bothnia. The dataset presented is very rich and deserves to be presented and published in The Cryosphere. The main topic of this paper is very important and really relevant.

Nonetheless, I feel like this paper has been rushed. Major editing (English especially) is needed to make the content more comprehensive especially for the Results, Discussion and Conclusion sections.

On the content, even though the dataset if very rich, I feel like the authors did not take the time to set up a proper methodology to exploit the full potential of the dataset. For example, the different EM dataset were not extracted in the same area each years (with their know issues in terms of data collection) and that makes it very difficult to make any

solid conclusions from the current methodology. I would have seen the EM data used to validate/correlate the information given in the ice charts and then used the charts to give some more solid conclusions and give a good discussion on the pros and cons of each dataset and how it affects the interpretation of the data from charts. This is just an example, I strongly advise that the methodology be revisited but the dataset presented definitely has the potential to give great results.

Another comment I would add is the literature review. Many studies have been published on sea ice thickness around the world (Arctic and Antarctic). It would be good for the authors to describe how their study compares to other studies made in the past. Also, some statements in the paper are not supported by any references, this has to be addressed.

I added comments in the attached pdf that should be addressed as well.

I recommend major reviews be done before another revision round.

Please also note the supplement to this comment:
https://www.the-cryosphere-discuss.net/tc-2018-87/tc-2018-87-RC2-supplement.pdf

[Figure]

**Supplement:**

[revised manuscript text omitted]

---

## Author Comment (AC1) · 5 Sep 2018

We would like to thank the referees for their thoughtful reviews and the time they have used to read our manuscript. We found the comments extremely valuable. As requested, especially by the referee #2, we have now made new analysis from the data and taken all other referee comments into consideration. We shall submit the revised, in our opinion much improved, manuscript promptly. Below are the referee comments in black and our responses in blue.

**Anonymous Referee #1**

The paper presents ice thickness data from EM measurements and drillings in the framework of level ice thickness indications in Baltic ice charts. The data are new and valuable and deserves publication.

However, the paper needs to be cleaned up. The language is often unclear (or poor English) and important information for the reader less familiar with the Bay of Bothnia is missing or hard to find.

There is a lack of discussion of uncertainty/error sources such as the sampling strategy. Not the same areas are sampled by the HEM each year but this is hardly discussed in the paper. This should feature more prominently in the discussion/conclusion sections.

We thank for these suggestions and we have done the following changes: We have rephrased many unclear sentences. Also, we expect the professional copy-editing of TC before final publication to correct remaining language problems. To make the important information easier to find we have added coordinates and place names to figure 1 and done a new table 1, where the information of the EM measurements is presented.

The HEM datasets have been measured for different research purposes where ice thickness has been needed, without any long term strategy. There has not been any year to year monitoring. This will be explained in the revised manuscript.

Specific comments:
P1L14: in a pure -> in pure
Changed

P1L16-30: A map with the place names that are referred to in the paper would be very helpful. Perhaps figure 1 could be used.
Place names and coordinates added to figure 1.

P2L2 and L15. These statements say almost the same but still are slightly contradictory. Does all ice grow thermodynamically in the fast ice zone, or is this only the case in more sheltered

areas. A follow up question is if the areas where the drillings and EM measurements are carried out can be considered 'more sheltered'?

Fast ice grows first in the more sheltered areas and there the ice grows only thermodynamically. During the ice season the fast ice zone expands to shallow (depth <10 m) drift ice area. The drillings are made near the coast, so at least most of them are carried out in 'more sheltered' areas. The EM measurements have been done in the drift ice zone. We have clarified the statements in the manuscript.

P3L4: What does 'the entire time series' men. Please give start and end years.

"the entire time series" replaced with start and end years

P3L8: This statement seems to be contradicted by the conclusions section. Perhaps it is more the interannual variability of ice extent that is related to NAO, and not the thickness. The term 'ice conditions' normally refers to ice extent AND thickness etc. I suggest to change ice conditions to ice extent here.

Ice conditions -> ice extent

P3L30: It depends… (what depends?) Paragraph should be rewritten.

The sentence rewritten "The ice thickness distribution and the amount of ridges affect navigation more."

P3L31: We are now using… (since when?) Do you mean 'In this study we use…'?

"We are now using" -> In this study we are using

P3L32-33: from fast ice zone -> from the fast ice zone

Changed

P4L1: Here I guess by conditions you mean thickness?

"ice conditions" -> ice thickness

P4L15: Refer to figures 1-3 for an outline of the area of interest.

Reference to figure 1 added

P4L26: satellite images and observations. Which observations??

Observations here are drillings at fixed stations near the coast and from icebreakers. In addition, visual observations are done by the crew of icebreakers. The sentence has been clarified.

P4L28: upturning -> turning

We think "tilting" is better word to replace upturning.

P5L5: of degree -> of a degree
Changed

P6L4: of approximately for HEM bird -> ??
"of approximately" removed as was suggested in the supplement of referee #2

Section 2.2, 2.2.1 and 2.2.2: There is some redundancy in the 3 EM sections. Perhaps these could be combined.
We have improved the sections so that there is less redundancy.

P7L20: solve -> investigate
Changed

P7L29: have been -> were
Changed

P8L5: not represent -> not fully represent
Changed

P8L2-5: Note that there is a cyclic feedback between FDD and ice extent (and thickness) in the sense that lots of FDDs leads to more and thicker ice, but also that lots of ice shelters the warm ocean from the atmosphere and thus leads to more FDDs.
This might be true for the Arctic, but for Bay of Bothnia heat flux from the sea is minimal compared to the large scale atmospheric transport of heat.

P8L8: The results… (results of what??, this is a bad way to start a sentence)
Sentence rewritten: We calculated the results from ice charts over 14 seasons 2003-2016.

P8L23: three fourths -> use percentage instead (e.g. 75% although from figure 4 it seems to be less than 75%)
"three fourths" -> 75% (from the pack ice 75% is rafted or ridged)

P9L1: The growth before this is linear. This may be true in the average sense, but most likely not in the individual years, so please be more specific.
We removed the whole sentence.

P9L11: data -> datasets
Changed

P9L18: Histograms of… A specification of the area of interest is needed here. (and perhaps a discussion of the sampling of this area each of the years)

We have extended the description in the text.

P9L18 (and fig 7): I notice negative ice thicknesses in the histograms. This needs a comment/explanation.
We have explained the negative ice thicknesses in the text.

P10L1: drift ice area west of hailuoto – show on a map (f.ex. figure 6)
Already shown in figure 6. Now added reference to text.

P10L12: have been -> were
Changed

P10L13: there is a lot -> there was a lot
Changed

P10L17: That is clearly -> This is clearly
Changed

P10L30-31: I suggest to rephrase to: severe winter 2003 where the EM data has its highest values and there was much (or a large fraction of) ridged ice.
Changed to: severe winter 2003 where the EM data has its highest values and there was much ridged ice.

P11L5-6: for the first time so that it has been reliably recorded ???
The sentence corrected to "for the first time so that it has been reliably recorded by satellites".

P11L6: that winter -> the 2015 winter
Changed

P11L11-12: maximum ice thickness in the fast ice zone in years 2003-2016 was over 0.8. So the max in each year was above 0.8m or in just one of the 14 years??
The maximum was above 0.8 m just one of the 14 years. Sentence changed, year (2010) and maximum value (0.86 m) added to the text.

P12L17-18: In a similar way…. - Please rewrite sentence to make more sense.
Sentence rewritten.

P12L23: but the reality is much more severe – Please rewrite. The level ice thickness given in the charts is also a part of the reality.
Sentence rewritten.

Note that the above specific language corrections are not the only ones needed for the paper to become acceptable.

We will send the manuscript to linguistic revision to correct the language.

**Anonymous Referee #2**

General Comments: This paper studies the evolution of ice thickness in the Bay of Bothnia. The dataset presented is very rich and deserves to be presented and published in The Cryosphere. The main topic of this paper is very important and really relevant.

Nonetheless, I feel like this paper has been rushed. Major editing (English especially) is needed to make the content more comprehensive especially for the Results, Discussion and Conclusion sections.

On the content, even though the dataset if very rich, I feel like the authors did not take the time to set up a proper methodology to exploit the full potential of the dataset. For example, the different EM dataset were not extracted in the same area each years (with their know issues in terms of data collection) and that makes it very difficult to make any solid conclusions from the current methodology. I would have seen the EM data used to validate/correlate the information given in the ice charts and then used the charts to give some more solid conclusions and give a good discussion on the pros and cons of each dataset and how it affects the interpretation of the data from charts. This is just an example, I strongly advise that the methodology be revisited but the dataset presented definitely has the potential to give great results.

Thank you for these suggestions. As authors we agree that the dataset at hand is important and should be published. We rethought our analysis of the dataset and agreed with the reviewer that further analysis indeed improved the manuscript significantly. Firstly, we analyzed the fraction of deformed ice. Secondly, we studied the impact of the coastal zone on ice thickness. We added Table 4 with fraction of thermodynamic and ridged ice in helicopter EM data and Figure 10 with ice thicknesses from helicopter EM and ice charts shown with distance from fast ice edge. We think this has improved the manuscript significantly and hope reviewers will agree.

The new text chapters based on this data are in section 4. As mentioned in the response to referee #1, we will send the manuscript to linguistic revision to correct the language.

Another comment I would add is the literature review. Many studies have been published on sea ice thickness around the world (Arctic and Antarctic). It would be good for the authors to describe how their study compares to other studies made in the past. Also, some statements in the paper are not supported by any references, this has to be addressed.

We have added a chapter to section 1 with references from studies carried out in the Arctic and Antarctic (Haas, 1998; Haas et al., 2008; Haas et al., 2010; Weissling et al., 2011; Meier et al., 2014; Haas and Howell, 2015) We have also added references to several statements that where without reference.

I added comments in the attached pdf that should be addressed as well.

I recommend major reviews be done before another revision round.

Here are the changes we have done based on the attached pdf:

P1L14 "can be even manyfold thicker" -> can be much thicker

P1L17 "extends" -> extending

P1L18 Comma added after Service

P1L20-21 "Comparable major seas that are ice covered seasonally" -> Other major seas that are seasonally covered

P1L25-26 Sentence rephrased to: "During the last 100 years the Bay of Bothnia has not frozen completely over only during extremely mild winters 2014/2015 for sure and most probably also in 1929/1930"
We mean here, that the bay usually freezes completely over, but not in these two years.

P1L30 Reference or data to "there is little exchange of ice with the Gulf of Bothnia in the south"
We haven't got any reference. Sentence rephrased.

P2L2 Comma added after zone

P2L4 Comma added after thickness

P2L5 Comma added after zone

P2L5&L11 We have added general reference

P2L18 "different kinds and ages" -> different thicknesses

P2L26 "an over" -> a more than

P2L32 "Also almost all" -> Also, most of the

P3L15 "Specifically it is not known," -> Specifically, it is not known

P3L19-21 Good point that the sentences should be moved to discussion. We moved them there.

P3L28 Comma added after winter

P3L28 Reference for statement "Every winter ships need assistance from ice breakers."
We added reference.

P3L30 "necessary" -> necessarily

P3L30 "to" -> for

P3L30 "of" -> on

P3L34-P4L1"We believe that the pooled data set from the campaigns is sufficiently extensive to reveal the main features of regional and interannual variation in ice conditions."

remove this sentence from introduction, it is too subjective "we believe". Also, if this comment is to be added further in the text I would specify that it is sufficient to reveal "modern" features since previously the authors mention the data set is not suited for climatology conclusions.
We have rephrased the sentence.

P4L7-8 References for different measurement techniques
References (Eicken, 2009; Haas, 2017) added.

P4L15 "we examined" removed

P4L17 "We indicate the winters so that for example 2003 means the winter 2002-2003."
Rephrased.

P4L19 "Finnish Meteorological Institute ice service" -> the ice service of the Finnish Meteorological Institute

P4L30 Is there any documentation from the Finnish Meteorological Institute that could be cited here to support or bring more details on how the charts are produced and how the thickness is estimated when no ship observations are made
Unfortunately, there is no documentation that would be valid.

P5L8 "and the latest ice chart was assumed to stay valid during the intermediate days." I would put a degree of uncertainty here. If both charts give the same information, it is safer to say the ice conditions are valid but if the two charts differ than the uncertainty is higher for the ice conditions.
We wrote more information to clarify the matter.

P5L11 I would like to see a table summarizing the data acquired, when (years), how (plateform), footprint, etc. A figure showing the Bay of Bothnia and the different locations of the datasets (colors could be used for different years) acquired would be greatly helpful to understand the study.
We added Table 1 with EM data years, dates, instrument etc. Figure 1 has now the place names and the different datasets are in figure 6.

P6L4 "of approximately" removed

P6L20 "at" -> with

P6L21 "than of" -> compared to

P7L22-24 Is there another weather station close to the study area with daily air temperatures? This is a major unknown for calculating FDDs. If not, I suggest using reanalysis data like ERA-Interim to get the daily mean surface air temperature.
We got the daily air temperature from the same Hailuoto Keskikylä station and calculated FDDs from daily data. We have updated the numbers in tables and in text.

P7L31-32 Put a reference for the website and when it was consulted.
Reference added.

P8L9-10 "In addition to that ice season is longer in the north than in the south, a clear tendency to longer season in more shallow coastal areas is seen." Needs editing
We have rephrased the sentence.

P8L16 "boundary" -> boundaries

P8L16 remove "there"
Sentence rephrased.

P8L16 "the" removed

P8L17 "The seasonal development of the decomposition of Bay of Bothnia ice area into ice types is in Fig. 4." Needs editing
Sentence rewritten.

P8L19-21 "The fast ice expands on the average to the mid-March, begins to decline rapidly in mid-April, and disappears at mid-May. That drift ice remains after fast ice has melted is a feature typical to the basin." Needs editing
We rephrased the sentences.

P11L2 "Our data shows that large interannual variability exists between ice seasons."
The ice charts maybe but the other datasets are not looking in the same area...
We have added more text to clarify the matter.

P11L21 "even manyfold" -> much

**References**

Eicken, H., Gradinger, R., Salganek, M., Shirasawa, K., Perovich, D., and Leppäranta, M. (Eds.): Field techniques for Sea-ice Research. University of Alaska Press, Fairbanks, 2009.

Haas, C.: Evaluation of ship-based electromagnetic inductive thickness measurements of summer sea-ice in the Bellingshausen and Amundsen Sea. Cold Regions Science and Technology, 27, 1–16, 1998.

Haas, C: Sea ice thickness distribution. In: Thomas, D. N. (Ed.), Sea ice, Wiley-Blackwell, 664 p, 2017.

Haas, C., and Howell, S. E. L.: Ice thickness in the Northwest Passage, Geophys. Res. Lett., 42, 7673–7680, doi:10.1002/2015GL065704, 2015.

Haas, C., Pfaffling, A., Hendricks, S., Rabenstein, L., Etienne, J. L. and Rigor, I.: Reduced ice thickness in Arctic Transpolar Drift favors rapid ice retreat. Geophysical Research Letters, 35, L17501, doi:10.1029/2008GL034457, 2008.

Haas, C., Hendricks, S., Eicken, H., and Herber A.: Synoptic airborne thickness surveys reveal state of Arctic sea ice cover, Geophys. Res. Lett., 37 ,doi:10.1029/2010GL042652, 2010.

Meier, W. N., Hovelsrud, G. K., Van Oort, B. E. H., Key, J. R., Kovacs, K. M., Michel, C., Haas, C., Granskog, M. A., Gerland, S., Perovich, D. K., Makshtas, A. and Reist, J. D.: Arctic sea ice

in transformation: a review of recent observed changes and impacts on biology and human activity. Reviews of Geophysics, doi: 10.1002/2013RG000431, 2014.

Weissling, B.P., Lewis, M. J., and Ackley, S. F.: Sea-ice thickness and mass at Ice Station Belgica, Bellingshausen Sea, Antarctica, Deep-Sea Research II, 58, 2011.